Effectiveness of topical antibiotics in treating corals affected by Stony Coral Tissue Loss Disease

Neely Karen L. kneely0@nova.edu
Macaulay Kevin A.
Hower Emily K.
Dobler Michelle A.
Halmos College of Natural Sciences and Oceanography, Nova Southeastern University , Dania Beach, FL , USA
Banaszak Anastazia
Electronic publication date: 2020 Jun 9
Publication date: 2020
Volume: 8
Electronic Location ID: e9289
Received 2020 Jan 10; Accepted 2020 May 13
Copyright: © 2020 Neely et al.
Copyright year: 2020
Copyright holder: Neely et al.
License: This is an open access article distributed under the terms of the Creative Commons Attribution License, which permits unrestricted use, distribution, reproduction and adaptation in any medium and for any purpose provided that it is properly attributed. For attribution, the original author(s), title, publication source (PeerJ) and either DOI or URL of the article must be cited.
License URL: https://creativecommons.org/licenses/by/4.0/

Keywords: Stony Coral Tissue Loss Disease, Antibiotic, Disease treatment, Florida Reef Tract

Funding: Florida Department of Environmental Protection’s Office of Resilience and Coastal Protection Funding for these activities was provided by the Florida Department of Environmental Protection’s Office of Resilience and Coastal Protection. The funders had no role in study design, data collection and analysis, decision to publish, or preparation of the manuscript.

==============================
Since 2014, Stony Coral Tissue Loss Disease (SCTLD) has led to mass mortality of the majority of hard coral species on the Florida Reef Tract. Following the successful treatment of SCTLD lesions on laboratory corals using water dosed with antibiotics, two topical pastes were developed as vehicles to directly apply antibiotic treatments to wild corals. These pastes were tested as placebos and with additions of amoxicillin on active SCTLD lesions on multiple coral species. The effectiveness of the pastes without antibiotics (placebo treatments) was 4% and 9%, no different from untreated controls. Adding amoxicillin to both pastes significantly increased effectiveness to 70% and 84%. Effectiveness with this method was seen across five different coral species, with success rates of the more effective paste ranging from 67% (Colpophyllia natans) to 90% (Orbicella faveolata and Montastraea cavernosa). Topical antibiotic application is a viable and effective tool for halting disease lesions on corals affected by SCTLD.

Introduction

Beginning in 2014, a disease since named Stony Coral Tissue Loss Disease (SCTLD) appeared on scleractinian corals near Miami, Florida (Precht et al., 2016). It has since spread throughout the Florida Reef Tract with a spatial pattern following a contagious model of transmission (Muller et al., 2020). Beginning in 2017, SCTLD began appearing in other regions of the Caribbean (Alvarez-Filip et al., 2019; Weil et al., 2019). The disease is known to affect over 20 species of corals and is characterized by multifocal acute lesions that in some cases are preceded by a bleaching margin (FKNMS/DEP, 2018). It is highly virulent, and capable of being transmitted by physical contact and also through seawater (Aeby et al., 2019). Progression of lesions across a colony is rapid compared to most other coral diseases, and in the majority of cases, lesions result in complete colony mortality. Ecosystem impacts are substantial and include significant decreases in coral cover, colony density and biodiversity (Precht et al., 2016; Walton, Hayes & Gilliam, 2018).

Fulfillment of Koch’s postulates for coral diseases is particularly difficult (Richardson, 1998) and definitive pathogen identification for SCTLD has not been successful. However, efforts to identify the cause of the epidemic are ongoing and have identified differences in bacterial communities between healthy and diseased corals (Meyer et al., 2019; Rosales et al., 2020). Additionally, early laboratory work noted that water dosing with antibiotics resulted in disease cessation (O’Neil, Neely & Patterson, 2018; Aeby et al., 2019), suggesting a bacterial component. Though both amoxicillin and ampicillin water baths were effective, ampicillin was preferred as it dissolved more easily. Follow-up efforts by NOAA’s Coral Disease and Health Consortium (C. Woodley, 2018–2020, personal communication) developed a modified dental paste that could be topically applied to disease margins; this product is still in use by laboratories and aquariums treating SCTLD-affected corals (O’Neil, Neely & Patterson, 2018). However, the use of the modified dental paste requires patting the coral dry and maintaining it in low water flow for 18 h, making it impracticable on wild corals. To resolve this, partnerships between the authors, the Florida Aquarium’s Center for Conservation, and a pharmaceutical formulation and manufacturing company (Ocean Alchemists LLC and CoreRx Pharmaceuticals) led to the development of topical pastes that could be used in field applications to determine whether SCTLD could be stopped on in situ colonies with active lesions.

This study compared the effectiveness of untreated controls, two placebo topical pastes (here termed Base 2b and New Base), and both pastes with amoxicillin additives to determine whether disease lesions could be halted.

Materials and Methods

Corals affected with SCTLD were selected for treatment at Sand Key (Fig. 1) in the lower Florida Keys (permit from Florida Keys National Marine Sanctuary #2019-115). Colonies were located within a 4,000 m2 area ranging in depth from 5 to 13 m.

Figure 1 Location of research site.

Map showing research site (star) at Sand Key, Florida Keys. Grey indicates land, pink represents patch reefs, and red indicates spur and groove reefs.

A total of 61 corals representing five species were selected for experimental treatment in October 2019. Each colony had between 1 and 12 active SCTLD lesions, and a total of 171 lesions were treated (Table 1). Due to the limited availability of suitable colonies and the varying number of lesions on each, the numbers of colonies and lesions were not the same across species and treatments. Three species (Colpophyllia natans, Montastraea cavernosa, Orbicella faveolata) were represented across all treatments. C. natans has been identified as a highly susceptible species, while M. cavernosa and O. faveolata have been classed as intermediately susceptible species (FKNMS/DEP, 2018). Two additional highly susceptible species, Diploria labyrinthiformes and Pseudodiploria strigosa, were compared between just two treatment types.

Table 1 Sample sizes for each treatment.

Number of colonies (A) and lesions (B) receiving each treatment (“Amoxi” represents addition of powdered amoxicillin). Four-letter species codes represent: Colpophyllia natans (CNAT), Montastraea cavernosa (MCAV), Orbicella faveolata (OFAV), Diploria labyrinthiformes (DLAB) and Pseudodiploria strigosa (PSTR).

		Control	New Base Placebo	New Base + Amoxi	Base 2b Placebo	Base 2b + Amoxi	
A	
#Colonies	CNAT	2	2	2	2	1	
MCAV	2	3	2	3	5	
OFAV	2	3	3	5	5	
DLAB	5	4	
PSTR	4	6	
B	
#Lesions	CNAT	8	2	7	8	3	
MCAV	4	6	4	9	9	
OFAV	4	19	12	14	23	
DLAB	8	8	
PSTR	8	15	

Selected colonies all had visibly active and rapidly progressing SCTLD disease lesions as identified by at least 1 cm of bright white bare skeleton adjacent to live tissue. Colonies had maximum linear dimensions ranging from 12 to 432 cm. Each colony was tagged and mapped for future identification. A masonry nail (2″) was hammered into each lesion to identify the location and progression of the disease margin.

Colonies were randomly assigned one of five treatments.

Control: colony was tagged and nails were affixed at the disease margin, but no treatment was applied.

“Base 2b” Placebo: a proprietary (CoreRx/Ocean Alchemists) silicone-based paste that included polymers to mimic coral mucus consistency was applied directly to the disease margin(s).

Base 2b + Amoxicillin: the silicone-based paste was hand mixed with powdered amoxicillin (sourced from Phytotechnology Laboratories. 98.1% purity) in an 8:1 (base:amoxicillin) by weight ratio. The paste included time-release products that regulated release of the amoxicillin over a 3-day time period.

“New Base” Placebo: a proprietary (CoreRx/Ocean Alchemists) biodegradable hydrophobic ointment designed to hold and release antibacterial compounds.

New Base + Amoxicillin: the New Base Placebo was mixed with powdered amoxicillin in an 8:1 by weight ratio. Release modifiers in the base facilitated amoxicillin release over 3 days.

Treatments were prepared less than 6 h before application by mixing amoxicillin into treatments 3 and 5, and by packing treatments 2–5 into 60cc catheter syringes. At each affected coral, a treatment was squeezed from the syringe and pressed by hand onto the length of the disease margin in a band approximately 1 cm wide. Half (0.5 cm) of the application overlaid and anchored on to the dead skeleton while 0.5 cm covered adjacent live tissue. If there were multiple lesions on a coral, they all received the same treatment. The amount of treatment product applied to each coral varied with the number and the size of lesions but averaged 12.3 mL (±11.2 mL SD) per treated coral. Among the 28 amoxicillin-treated colonies, an average of 1.6 g (±1.7 g SD) of amoxicillin was applied for a total application of 39.6 grams at the site.

Corals were monitored 4 weeks after the initial treatment. At each coral, the number of effective and ineffective treatments were tallied. Photographs were also taken and arranged so that before and after photos of each lesion could be compared (representative samples: Fig. 2). All analyses were based on the photographic comparisons because more lesions could be positively identified. Effectiveness was defined as the cessation of disease progression at the treatment line. Ineffectiveness was defined as the lesion continuing unimpeded across the colony. After the 1-month monitoring, all active lesions on all surviving corals were treated with Base 2b + Amoxicillin. A total of 55 retreated lesions (5 C. natans, 19 M. cavernosa and 31 O. faveolata) were reassessed 2 months later for effectiveness.

Figure 2 Examples of treated SCTLD lesions.

Representative photos of placebo treatments and amoxicillin treatments. Species codes are: PSTR, Pseudodiploria strigosa; DLAB, Diploria labyrinthiformes; OFAV, Orbicella faveolata. Photos show the corals immediately before treatment was applied, immediately after treatment was applied, and 1 month after treatment. The exception is the “1 month” photo of the placebo DLAB (*), which was taken 2 weeks after treatment; the colony was totally dead at 1 month.

The proportion of halted lesions was compared across treatments using Fisher’s exact test (α = 0.05), which is suitable for unequal as well as small sample sizes. Lesions on the same coral were considered independent because: Microbiome studies of SCTLD colonies identify healthy regions of tissue adjacent to diseased regions (Meyer et al., 2019).

Field observations of lesion development on individual colonies over time note asynchronous appearance, suggesting independent development.

Additional analyses that do not assume lesion independence were conducted by comparing the proportion of halted lesions on each colony across species and treatments. Sample sizes were small, (between one and six colonies per species per treatment), unequal, and did not always pass Shapiro-Wilk normality or Brown-Forsythe equal variance tests. Therefore, comparisons were conducted using t-tests when assumptions were met and Mann-Whitney Rank Sum Tests when they were not.

Results

The percentage of effective lesion treatments varied by treatment type and, to some extent, species (Fig. 3). Across all species, 0% of the control (untreated) lesions halted. Overall effectiveness of New Base Placebo and Base 2b Placebo treatments on lesions were 4% (1/27) and 9% (4/47) respectively. When amoxicillin was added, effectiveness increased to 70% (16/23) for the New Base and 84% (49/58) for the Base 2b.

Figure 3 Effectiveness of treatments in halting disease lesions.

Number of halted (A) and active (B) lesions for each treatment type 1 month after treatments. Colors/patterns represent different species. Total percentage of lesions that halted under each treatment regime are shown above the halted lesion bars.

Because the treatments varied in the number of lesions on different species, analyses were further broken down by species (Table 2). C. natans treatments exhibited the least difference between placebo and amoxicillin treatments; when amoxicillin was added to both the New Base and the Base 2b, effectiveness increased by 29%. On O. faveolata, effectiveness increased 78% when amoxicillin was added to the New Base and 91% when added to the Base 2b. On M. cavernosa, the addition of amoxicillin increased effectiveness by 100% in New Base and 89% in Base 2b as compared to their placebo counterparts. Because of species rarity, D. labyrinthiformes and P. strigosa lesions treatments were restricted to Base 2b Placebo and Base 2b + Amoxicillin treatments; effectiveness was 0% with placebo treatments, while 88% (7/8: D. labyrinthiformes) and 73% (11/15: P. strigosa) of amoxicillin treated lesions were effective.

Table 2 Statistical comparisons of lesion treatment effectiveness.

Number of effective and ineffective lesion treatments for each species and treatment type and p-values from Fisher’s exact test comparisons between treatments for each species. Statistically significant results are highlighted in green.

	Control	New Base Placebo	New Base + Amoxi	Base 2b Placebo	Base 2b + Amoxi	
CNAT	
Effective: ineffective	0:8 (0%)	0:2 (0%)	2:5 (29%)	3:5 (38%)	2:1 (67%)	
Control	–	N/A (both zero)	0.2	0.2	0.055	
New Base Placebo	N/A (both zero)	–	1	1	0.4	
New Base + Amoxi	0.2	1	–	1	0.5	
Base 2b Placebo	0.2	1	1	–	0.54	
Base 2b + Amoxi	0.055	0.4	0.5	0.54	–	
OFAV	
Effective: ineffective	0:4 (0%)	1:18 (5%)	10:2 (83%)	0:14 (0%)	21:2 (91%)	
Control	–	1	0.008	N/A (both zero)	<0.001	
New Base Placebo	1	–	<0.001	1	<0.001	
New Base + Amoxi	0.008	<0.001	–	<0.001	0.594	
Base 2b Placebo	N/A (both zero)	1	<0.001	–	<0.001	
Base 2b + Amoxi	<0.001	<0.001	0.594	<0.001	–	
MCAV	
Effective: ineffective	0:4 (0%)	0:6 (0%)	4:0 (100%)	0:9 (0%)	8:1 (89%)	
Control	–	N/A (both zero)	0.029	N/A (both zero)	0.007	
New Base Placebo	N/A (both zero)	–	0.005	N/A (both zero)	0.001	
New Base + Amoxi	0.029	0.005	–	0.001	1	
Base 2b Placebo	N/A (both zero)	N/A (both zero)	0.001	–	<0.001	
Base 2b + Amoxi	0.007	0.001	1	<0.001	–	
DLAB	
Effective: ineffective	–	–	–	1:7 (13%)	7:1 (88%)	
Base 2b Placebo	–	–	–	–	0.001	
PSTR	
Effective: ineffective	–	–	–	0:8 (0%)	11:4 (73%)	
Base 2b Placebo	–	–	–	–	0.001	

Fisher’s exact tests identified similarities and differences among treatments (Table 2). On O. faveolata, M. cavernosa, and C. natans, there were no significant differences between untreated controls, New Base Placebo and Base 2b Placebo. On C. natans, there were no significant differences between any treatments (controls, placebos and non-placebos), although the p-value between untreated controls and Base 2b + Amoxicillin was 0.055. On both O. faveolata and M. cavernosa, there were significant differences between controls and both amoxicillin products. There were also significant differences between both placebo bases and their amoxicillin counterparts. There were no significant differences in effectiveness between New Base + Amoxicillin and Base 2b + Amoxicillin. On D. labyrinthiformes and P. strigosa, effectiveness of Base 2b + Amoxicillin was significantly higher than the Base 2b Placebo (p = 0.001). Across five of the six tested species, the percentage of lesions halted using both amoxicillin bases was between 73% and 90%. However, amoxicillin treatments on C. natans were less effective, particularly with the New Base (29%).

Treatments were also analyzed at a colony level in consideration that host genotype may play a role in treatment effectiveness. The percentage of halted lesions on each colony was compared across treatments for each species (Fig. 4). For all species, the percentage of halted lesions was higher for New Base + Amoxicillin or Base 2b + Amoxicillin treatments than for placebo-treated colonies and controls. Statistically, ANOVA on Rank tests across all treatment types had extremely low power and could not detect post-hoc differences between treatments by species (Table 3). However, individual comparisons did detect significant differences on P. strigosa between Base 2b + Amoxicillin and Base 2b Placebo (Mann-Whitney Rank Sum Test: p = 0.038), and also on O. faveolata between Base 2b + Amoxicillin compared to both Base 2b Placebo and New Base Placebo (Mann-Whitney Rank Sum Test: p = 0.008 and t-test: p < 0.001 respectively).

Figure 4 Percentage of effective lesion treatments per colony.

The average percentage of lesion treatments that were effective on each coral colony, separated by species and treatment type. Error bars indicate standard error.

Table 3 Statistical comparisons of the percentage of lesions halted on each coral colony.

Average percentage (± standard error) of halted lesions on each coral colony by treatment are presented for each species. Comparisons for each treatment show p-values from two-tailed t-tests where normality and equal variance assumptions were met, and from Mann-Whitney Rank Sum Tests where they were not. Statistically significant results are highlighted in green.

	Control	New Base Placebo	New Base + Amoxi	Base 2b Placebo	Base 2b + Amoxi	
CNAT	
Average % (±SE)	0 ± 0	0 ± 0	58 ± 42	21 ± 21	67 ± N/A	
Control	–	1	–	0.667	N/A	
New Base Placebo	1	–	0.333	0.667	N/A	
New Base + Amoxi	0.333	0.333	–	0.667	N/A	
Base 2b Placebo	0.667	0.667	0.667	–	N/A	
Base 2b + Amoxi	N/A	N/A	N/A	N/A	–	
OFAV	
Average % (±SE)	0 ± 0	3 ± 3	67 ± 33	0 ± 0	90 ± 6	
Control	–	0.495	0.136	1	0.095	
New Base Placebo	0.495	–	0.24	0.57	<0.001	
New Base + Amoxi	0.136	0.24	–	0.143	0.399	
Base 2b Placebo	1	0.57	0.143	–	0.009	
Base 2b + Amoxi	0.095	<0.001	0.399	0.009	–	
MCAV	
Average % (±SE)	0 ± 0	0 ± 0	100 ± 0	0 ± 0	80 ± 20	
Control	–	1	0.333	1	0.19	
New Base Placebo	1	–	0.33	1	0.19	
New Base + Amoxi	0.333	0.33	–	0.2	0.857	
Base 2b Placebo	1	1	0.2	–	0.07	
Base 2b + Amoxi	0.19	0.19	0.857	0.07	–	
DLAB	
Average % (±SE)	–	–	–	20 ± 20	75 ± 25	
Base 2b Placebo	–	–	–	–	0.135	
PSTR	
Average % (±SE)	–	–	–	0 ± 0	81 ± 16	
Base 2b Placebo	–	–	–	–	0.038	

One month after treatment, one of the control colonies and six of the Base 2b Placebo colonies had died completely. Of the surviving colonies, new lesions had developed on: 40% of the controls, 50% of the New Base placebo, 29% of the Base2b placebo, 29% of the New Base + Amoxicillin and 33% of the Base2b + Amoxicillin colonies. Failed and new lesions from all surviving colonies were treated with Base 2b + Amoxicillin. Fifty-five of those retreatments were re-surveyed after 2 months; 80% of C. natans lesions, 77% of O. faveolata lesions and 58% of M. cavernosa lesions had halted.

Discussion

Past uses of antibiotics on diseased corals have included utilization as a diagnostic tool to help identify bacteria as a presumptive pathogen in white band disease (Kline & Vollmer, 2011; Sweet, Croquer & Bythell, 2014) as well as water dosing to halt SCTLD lesion progression in laboratory and aquarium work (C. Woodley, K. O’Neil & C. Lewis, 2018–2020, personal communication). The use of skin wound treatment patches containing antiseptics, antioxidants and/or antibiotics have also shown promise in helping mechanically damaged corals to heal (Contardi et al., 2020). However, the results presented here represent the first known use of topical antibiotics as a disease treatment tool to preserve wild populations. Effectiveness of placebos was no greater than for untreated controls, but the addition of amoxicillin significantly increased the percentage of lesions halted.

Antibiotic application was successful at halting lesions on all tested species, but effectiveness on C. natans was lower, particularly for the New Base + Amoxicillin treatment. We suggest that the deep polyp grooves of C. natans create gaps where the treatment is not in contact with coral tissue. Divers noted that the New Base had poorer adherence to the coral colonies than the Base 2b, which may have been particularly pronounced in the highly rugose C. natans, thus creating ineffective treatment barriers. Careful application into grooves to ensure physical contact with the coral tissue is suggested.

Antibiotic effectiveness is likely to remain localized within the region of application rather than spreading throughout the colony; this was evidenced by the appearance of new lesions on some amoxicillin-treated corals. To minimize mortality, coral colonies required a 1-month revisitation in order to retreat any failed margins and to treat any new lesions. Longer-term studies are recommended and currently underway to determine appropriate visitation intervals and long-term maintenance requirements. Physiological studies to determine the spread of the amoxicillin throughout the colony and the mechanism for effectiveness are also recommended.

Treatment of SCTLD-affected colonies using topical amoxicillin paste is an option for SCTLD disease intervention. The methodology has already been utilized to save laboratory and aquaria corals (Florida Aquarium’s Center for Conservation, Keys Marine Laboratory, Frost Museum of Science), in-water nursery corals (Coral Restoration Foundation and Florida Keys National Marine Sanctuary), corals targeted for assisted reproduction efforts (Mote Marine Laboratory), and the preservation of over 2,000 large reef-building corals on the Florida Reef Tract (Nova Southeastern University, Harbor Branch Oceanographic Institute). Such actions have been and should continue to be weighed in a risk management scenario that considers unknown factors such as impacts on the treated corals’ microbiomes as well as potential antibiotic resistance. Though requiring an investment of time and resources for initial treatment and monitoring, topical antibiotic treatment is a viable tool for preserving high-value corals.

Conclusions

Topical amoxicillin treatments successfully arrested disease lesion progression on multiple species of stony corals affected by SCTLD. As this disease spreads throughout currently affected reefs as well as new regions of the Caribbean, this type of in-water intervention is an option to be considered within management response strategies. Follow-up studies on the physiological mechanisms, potential risks and long-term trajectory of treated corals are recommended.

Supplemental Information

Supplemental Information 1 Raw data of effective and ineffective lesion numbers across treatments on each colony at 1 month monitoring.

Click here for additional data file.

We are grateful to the editor and three reviewers for improvements to this manuscript.

Additional Information and Declarations

Competing Interests

Author Contributions

Field Study Permissions

Data Availability

The authors declare that they have no competing interests.

Karen L. Neely conceived and designed the experiments, performed the experiments, analyzed the data, prepared figures and/or tables, authored or reviewed drafts of the paper, and approved the final draft.

Kevin A. Macaulay performed the experiments, authored or reviewed drafts of the paper, and approved the final draft.

Emily K. Hower performed the experiments, authored or reviewed drafts of the paper, and approved the final draft.

Michelle A. Dobler performed the experiments, authored or reviewed drafts of the paper, and approved the final draft.

The following information was supplied relating to field study approvals (i.e., approving body and any reference numbers):

Experiment was permitted by Florida Keys National Marine Sanctuary #2019-115. Approval of antibiotics also authorized by the US Food and Drug Administration’s Office of Minor Use & Minor Species (FDA–OMUMS).

The following information was supplied regarding data availability:

Raw tallies of effective and ineffective lesions on each coral at 1 month are available in File S1.

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
