# Peer review of "Effectiveness of topical antibiotics in treating corals affected by Stony Coral Tissue Loss Disease"

_PeerJ, doi:10.7717/peerj.9289_

## Round 0.1 · original submission · Major Revisions

SCTLD is a worrisome syndrome that is affecting approximately 23 species of corals in the Caribbean. Beginning in 2014 in Florida, the disease has now spread down to the MesoAmerican Reef and has been detected in coral reefs on several Caribbean islands. This is the context of the research presented in this manuscript. Therefore, any possible treatment is likely to cause major interest and as such the manuscript is timely. However, the reviewers raise important issues that need to be taken care of and I also have some suggestions that need to be considered.

The major issue is that this report lacks scientific rigor and that needs to be remedied. The introduction needs to be improved by including previous publications on this topic and leading the reader to questions and/or hypothesis; the methods are unclear (e.g. which species were used, how many replicates of each species), no clear information is given on the statistics that were used (especially due to uneven distribution of replicates) and results need to be presented more clearly (for example in the statistics, 5 species are compared but apparently 7 were used; are you treating each lesion as being independent? Are you treating different species together or separately?), the discussion needs to be more objective in discussing the results and including previous research.
Another major issue is that from the outset there is an assumption that this is a syndrome caused by bacteria, however, is that supported by the literature? Or do you have supporting information? It would be a good idea to include that.

Another major question is why treat with antibiotics? Why amoxicillin? What is the reasoning behind choosing this treatment? More information should be given about the difference between base 2b and new base because there was some apparent improvement in effectiveness just due to their application as a placebo, suggesting that it is not completely a bacterial issue?

Do you have follow-up data after 4 weeks? I think it would be useful, even though I understand the difficult of doing this extra field work, because the implications of this study are so great. In particular, I do know that quite a number of scientists are concerned with the broad-band application of antibiotics in the field as a treatment for SCTLD. How does this affect the coral microbiome as well as the microbiomes in the water, in the sediment and on other species. Isn´t there a risk of promoting antibiotic resistant strains?

In terms of the analysis of the data, from what I understand your design used 64 colonies and 5 treatments. However the number of “treatments” for control were 21 (according to the text but according to Fig 1 bottom it is less than 20 and needs to be resolved), for placebo new base 27 for placebo new base with amoxicillin 23 for base 2b placebo 47 and for base 2b placebo with amoxicillin 63. These are not treatments but replicates, right? Or more specifically, number of lesions that were “treated” and they add up to 181. So a number of important questions arise: how many replicates of each coral species underwent the different treatments?

Also, if a coral colony has more than one lesion, treating it (statistically) as an independent lesion is incorrect, right? It sounds like psuedoreplication and I would consult a statistician. I believe what needs to be done is to use an average for each colony when it has more than one lesion. Ie you should have 64 data points not over 180.

In the figures: axes read # treated lesions but I would change to number of lesions as the control was not treated. # of effective lesions sounds odd (you are testing the antibiotic for effectiveness not the lesions), I suggest Number of lesions responding to treatment etc. Same for ineffective lesions.

·

Basic reporting

I’m surprised to see a lack of references to studies that have used antibiotics on corals before which I’m sure gave these authors the ideas and concepts of this study in the first place
Sweet, M. J., Croquer, A., & Bythell, J. C. (2014). Experimental antibiotic treatment identifies potential pathogens of white band disease in the endangered Caribbean coral Acropora cervicornis. Proceedings of the Royal Society B: Biological Sciences, 281(1788).
Sweet, M., and John C. Bythell. White Syndrome in Acropora muricata: Non-specific bacterial infection and ciliate histophagy. Molecular Ecology 24.4 (2015):

Experimental design

Can the authors allude to the permits given for such experimental work in situ in the text – its noted this is written at the bottom but would be good within text.
Were the diseased colonies tagged and monitored beforehand to ensure they were progressing?

Validity of the findings

L106 present significant results in text
Photos of each colony are mentioned – this would be valuable information to provide and put into a composite figure showing the various success rates with scales

Additional comments

See above

·

Basic reporting

See below

Experimental design

See below

Validity of the findings

See below

Additional comments

Effectiveness of topical antibiotics in treating corals affected by Stony Coral Tissue Loss Disease by Karen L Neely Corresp., 1 , Kevin A Macaulay 1 , Emily K Hower 1 , Michelle A Dobler PEERJ-44795

Overview: Authors treated a series of colonies of multiple species of corals with 4 formulations of a paste ladened or not with amoxicillin and gauged efficacy of this treatment to stop progression of tissue loss in corals affected with SCTLD. They found ca. 29-86% of corals with amoxicillin-laden paste to have halted lesion progression whereas placebo paste success ranged from 4-7%. There was a ca.50-60% recurrence rate a month after treatment. Authors are to be commended for trying to find a solution to the pressing problem of SCTLD in FL, but it is unclear whether perceived benefits outweigh broader risks. Several points to consider: 1) No convincing link has been made between bacterial infections and SCTLD at the cellular level. Indeed, in the study cited (Aeby et al. 2019), all they showed was transmission, but many things can look transmissible (e.g. autotomy in sea stars (Motokawa 2011)) but are not necessarily caused by infectious agents. No efforts were made in that study to look at the corals on microscopy to see if bacteria were truly the cause of experimental tissue loss. 2) Treating corals with antibiotics seems a bit risky and somewhat irresponsible given that antibiotics can affect other organisms on the reef. Moreover, with the pervasive problems with antibiotic resistance, spreading more antibiotics in the environment, particularly to treat something that does not deal with the root of the problem, could aggravate things in the longer term. Why did the authors not consider alternatives such as a paste laden with bleach (Beurmann et al 2017) which would be just as efficacious in killing all manner of microbiota but not engender resistance? 3) Treatments such as these also introduce a moral hazard. Notwithstanding the practicality of treating 10s of km^2 using this technique and dumping huge amounts of antibiotics in the environment in the process, such solutions allow decisions makers to detract from more fundamental changes in coastal management that need to happen to save Florida's Coral reefs. The Florida reef tract suffered its first tectonic change in coral cover with the loss of Acropora in the 1980s (Aronson & Precht 2001), and now we are seeing a second wave of collapse with remnant species. It is notable that this ongoing SCTLD started in 2014 near the Miami Channel that was being dredged releasing a huge amount of sediments (Miller et al. 2016; Barnes et al. 2015) during a time of unusually high temperatures which themselves were stressful to corals (Precht et al., 2016). Notably, to expedite the dredging process, the State of Florida imposed strict limits on resource agencies' ability to evaluate or mitigate the sediment damage of dredging activities in order to move things along (Miller et al. 2016). It is marshalling the political will to curb practices such as these that will have the most beneficial long term value to save Florida's corals. Some more specifics:

Lines 87-88: How did those particular colonies fare? Was the treatment just as efficacious as what you found after the one month period?

Methods: Can you provide an exact formulation for the paste? or is it proprietary?

line 59: What defines small?

Lines 111-113 suggests that the amoxicillin is not preventing recurrence of disease. Moreover, there is the implicit assumption here that bacteria are the problem (you are treating with antibiotics and lesions stop). But it could just be that bacteria are aggravating an underlying problem, and it seems that efforts should be going to sorting what exactly is causing SCTLD in the first place.

Lines 124-125: Perhaps useful in aquaria but do you really think this tool is a viable management option given the need to reapply multiple times?

Lines 126-128: Disagree. Clearly, your method does not stop new lesions from forming, so applying to high value corals does not eliminate their exposure to cause of SCTLD. For those "rare or high value corals", the best bet will be to remove them from habitat and hold them in aquaria.

References
Aeby, Greta S., Blake Ushijima, Justin E. Campbell, Scott Jones, Gareth J. Williams, Julie L. Meyer, Claudia Häse, and Valerie J. Paul. 2019. 'Pathogenesis of a Tissue Loss Disease Affecting Multiple Species of Corals Along the Florida Reef Tract', Frontiers in Marine Science, 6.
Aronson, R. B., and W. F. Precht. 2001. 'White-band disease and the changing face of Caribbean coral reefs', Hydrobiologia, 460: 25-38.
Barnes, B.B., C Hu, C Kovach, and R.N. Silverstein. 2015. 'Sediment plumes induced by the Port of Miami dredging: Analysis and interpretation using Landsat and MODIS data', Remote Sensing of Environment, 170: 328–39.
Beurmann, S., C. M. Runyon, P. Videau, S. M. Callahan, and G. S. Aeby. 2017. 'Assessment of disease lesion removal as a method to control chronic Montipora white syndrome', Dis Aquat Organ, 123: 173-79.
Miller, M.W., J Karazsia, C.E. Groves, S Griffin, T Moore, P Wilber, and K Gregg. 2016. 'Detecting sedimentation impacts to coral reefs resulting from dredging the Port of Miami, Florida USA', PeerJ, 4: e2711.
Motokawa, T. 2011. 'Mechanical mutability in connective tissue of starfish body wall', Biological Bulletin, 221: 280–89.
Precht, W.F., B.E. Gintert, M.L Robbart, R Fura, and R van Woesik. 2016. 'Unprecedented Disease-Related Coral Mortality in Southeastern Florida', Scientific Reports, 6: 31374.

·

Basic reporting

This is an important contribution to the field that presents the most recent advances in treatment to a deadly bacterial-induced disease in foundational scleractinians. I recommend however, that the authors refine and expand a bit the introduction and
experimental design with the treatment methodologies used for future successful replication. The manuscript is well written, but the introduction (and references) could be expanded a bit, specially by comparing with the other deadly disease (WPD) that has very similar signs to SCTLD, and has been around for decades. Many authors believe this new one is just another variation of the original WPD, so emphasizing the main differences would be a good start.

Overall structure, figures tables are fine. There are however, no clear stated questions and hypotheses, they are obvious, but maybe, to follow the manuscripts' professional structure, these should be included at the end of the introduction.

Experimental design

Most results are presented in the context of success or failure of treatments with regards to lesions within colonies of different coral species. However, up to the point of reading the results, the reader has no idea how many colonies/lesions in how many different species were used, or if the 5 step design was applied consistently to the seven different species used (balanced design).

From Fig.1 it is obvious that this is not the case. It would then be advisable that authors expand and clarify the experimental design and how the analysis was done. Number of lesions treated by colonies/species, variable (s) used (lesions/colony/or species), conditions for the Z-tests (proportions) used, etc. This not clearly stated in the methods. Maybe a table or diagram with the experimental design including species/colonies/ lesions and the treatments used could clarify this.

Validity of the findings

Discussion should present more detail with regards to what were the main differences across the different species, and treatments. Which species showed higher effectiveness and maybe, some explanation/speculation of why. Is there any phylogenetic relationship amongst those that responded better, or failed? Any considerations about how the antibiotic could affect the natural microbiome of the different species? Any suggestions to improve the efficiency of the methods in the field? These are important topics that could be included in the discussion and conclusions.

Additional comments

I would like to congratulate and thank the authors for taking on this important aspect of the disease problem impacting coral reefs Worldwide. This is an important contribution to the field that will be widely used and cited in the near future. However, I think that with a little more of work (see general and specific comments), it would be greatly improved . I recommend that the authors refine and expand a bit the introduction, the experimental design and the field methods, and complement the discussion and conclusions as indicated in the general and specific comments.
General and specific comments:

Abstract

Line 23: …….overall effectiveness…
Lines 24-25: but, how many of the total susceptible spp? ….and effectiveness was not the same for all species, right?? Clarify in tis sentence.
Line 25: as the conclusion, emphasize that this is by far the most effective method to treat SCTLD diseased colonies today.

Introduction:

Lines 51-57: Please clarify if colonies selected belong to just one species, the most susceptible species or species with varying susceptibility. How many colonies of each of the different species were selected? This has bearings on the following experimental design. If the species was not considered as a factor, please explain why. It would be a confounding factor in the results and interpretation.

Lines 66-72: What is the difference between Base B and the New Base??
Line 78: ……..0.5 cm covered infected live tissue.

Any considerations about how the antibiotic could affect the natural microbiome of the different species? Could it eliminate beneficial bacteria to the host?

Materials and Methods:

Please include a figure/map with the location of the experimental area


Results

Most results are presented in the context of success or failure of treatments with regards to lesions within colonies of different coral species. However, up to this point, the reader has no idea how many colonies/lesions in how many different species were used, or if the 5 step design was applied consistently to the seven different species used (balanced design).

From Fig.1 it is obvious that this is not the case. It would then be advisable that authors expand and clarify the experimental design and how the analysis was done. Number of lesions treated by colonies/species, variable (s) used (lesions/colony/or species), conditions for the Z-tests (proportions) used, etc. This not clearly stated in the methods. Maybe a table or diagram with the experimental design including species/colonies/ lesions and the treatments used could clarify this.

Results for seven different species are presented in figure 1, but two of these spp. are never mentioned in the results and discussion?

Line 101 -103: This summary sentence must be more specific: … Overall, Base 2B + amoxicillin effectiveness was 86% (54/63) for all colonies treated (all species), with P. strigosa showing the highest % of failures.

An effectiveness of 73% for example, means that 11 colonies showed arresting disease signs from a total of 15 colonies treated (base + amoxicillin) right? Those 15 do not include controls and placebos, right? Maybe adding “antibiotic treated”… before …..colonies….. in the results would clarify this. Results and discussion are a bit confusing because authors mix results for the total number of colonies (or lesions), and those for individual species (i.e. Lines 99-103; 106-108; 111-113; etc.).
.
Discussion:

Line 120: …. The addition of…….. This should be in the results.


Discussion should present more detail with regards to what were the main differences across the different species, and treatments. Which species showed higher effectiveness and maybe, some explanation/speculation of why. Is there any phylogenetic relationship amongst those that responded better, or failed? Any considerations about how the antibiotic could affect the natural microbiome of the different species? Any suggestions to improve the logistics and efficiency of the methods in the field? These are important topics that could be included in the discussion and conclusions that would improve the manuscript.

---

## Round 0.2 · Minor Revisions

Three expert reviewers have evaluated your resubmitted manuscript and their comments can be seen below. As you can see, all have suggestions on improvements to the manuscript.

·

Basic reporting

I still think this is a very basic study and not really publishable as a full paper but that in my opinion is the editors choice.

The text would also benefit from some stringent re writing for example L39 and L41 both start with the lead, However and this is only one example among many where sentence structure could be improved.

I also have a major worry about how or why antibiotics were used in such away in the first place, but as permits were given for this experiment again this is not really my place to say this, although I imagine the study will come under attack as to bad practice to trial such 'treatments' in the field.

L98 can you give some examples of the before and after photos highlighting successful and non-successful applications?

In the results where you talk about significant or not differences please put in the test and the value in brackets for example but throughout L139

beginning of discussion you might want to reference this new paper which explores a similar treatment albeit for a different coral disease in a different region https://www.nature.com/articles/s41598-020-57980-1

The discussion (as written) isn't really a discussion, more a report on what is advised for people to utilise this method of treating corals.

Experimental design

see above

Validity of the findings

see above

Additional comments

see above

·

Basic reporting

See below

Experimental design

See below

Validity of the findings

See below

Additional comments

Overview: This is a revision of an earlier version. There is no question that antibiotics and paste alleviate progression of lesions in corals to variable extent. Moreover, it appears that this method is already being deployed on a larger scale thus resulting in applications of large amounts of antibiotics on coral reef ecosystems absent any evident effort to gauge potential risks that such an approach may have on non-target reef organisms. Clearly the horse has left the barn. The implication is that SCTLD is caused by bacteria, and a solution is large scale application of antibiotics to coral reef ecosystems absent any sort of risk assessment. It only remains to give this approach and message scientific imprimatur. Given that the authors have done a reasonably good job to address most comments, and that they have shown unquestionably that amoxicillin does slow lesion development, I can only offer some additional comments to clarify things and perhaps provide a more balanced perspective.

Line 40: Pls reword "...definitive cause of SCTLD remains unknown". Until someone can show actual evidence of a pathogen at the cellular level, we can't say this disease is caused by an infectious agent.

Line 84: Pls indicate this is a proprietary ointment.

Lines 89-94: It would be useful here to know about how much material total was used to treat these corals. Please also indicate the source and manufacturer of the amoxicillin.

Also, the amounts you are using are not trivial. For instance, assuming 60cc/coral, you are dumping into the environment about 7.5 g of amoxicillin, so for 61 corals, you contributed about 450 g of antibiotics to the reef. You later go on to state that you treated 2000 corals with this formulation. That's basically dumping 15 kg of pure amoxicillin on a reef ecosystem. Even at half that, it would be 7 kg. Any idea of the potential ramifications of this? Yes, it degrades rapidly in the environment, but there are concerns about antibiotic resistance (1). There is evidence that amoxicillin can disrupt photosynthesis of algae (2). There is also the issue that the breakdown products of this compound could be more toxic than the parent (3-4). Perhaps a more balanced approach would be to acknowledge that this approach does carry some risks (perhaps citing some of the papers aforementioned). Indeed, it is very surprising that large scale efforts have been carried out in absence of any rational risk assessment as to potential side effects of these activities.

You also state in your rebuttal that you tried bleach and had little success. This seems really odd, because bleach is a very effective killer of bacteria, again bringing into question exactly what the amoxicillin is actually doing to slow lesions.

Lines 104-09: Again, you are assuming SCTLD is infectious, but the only evidence for this is field signs that can be very misleading (see my previous review). Please delete any reference to this being infectious as this could mislead readers.

Lines 177-178: What percent of treated corals by colonies developed new lesions? This would be interesting to present in results, simply as a sentence (e.g. "Follow up surveys revealed X, Y, and Z% of treated CN, OF, ML, etc. had new lesions).

Line 181: THere needs to be a mention here somewhere about how many Kg of amoxicillin you are planning or projecting to apply to reefs in the longer term. This would add transparency to those pondering potential side effects.

Lines 188-189: "Such actions have been and should continue to be weighed in a risk management scenario that considers unknown factors such as impacts on the treated corals’ microbiomes as well as potential antibiotic resistance." Precisely. Although there is lip service to the effect, I get the sense here that no actual risk assessment was made at all. I could almost understand going ahead with large scale treatments using, say, something biodegradable like bleach but you are here using a bioactive compound that degrades to potential toxic byproducts (see citations above). It is amazing that this has been allowed to go forward on a large scale, but I guess I see the rationale that "Something needs to be done" in spite of the possibility that the cure could be worse than the disease.

Figure 2. Caption Bottom is confusing. How can you have a negative number of lesions? Is this a percent decrease? Suggest delete Figure 2 and leave Fig 3 which is far more informative and essentially communicates the same thing. Simplifies presentation.

References:

1) https://www.intechopen.com/books/environmental-health-risk-hazardous-factors-to-living-species/amoxicillin-in-the-aquatic-environment-its-fate-and-environmental-risk
2) https://www.sciencedirect.com/science/article/abs/pii/S0166445X08002075
3) https://www.sciencedirect.com/science/article/abs/pii/S0166445X08002075
4) Fatta-Kassinos, D., Kalavrouziotis, I.K., Koukoulakis, P., Vasquez, M.I., 2011a. The
risks associated with wastewater reuse and xenobiotics in the agroecological
environment. Sci. Total Environ. 409, 3555–3563.

·

Basic reporting

This is an important contribution that provides potential mitigation treatments for the SCTLD epizootic problem and potentially, for future bacteria-caused diseases in marine invertebrates. I praise the authors for taking on the challenge, working with disease in corals is not easy task.
Authors addressed all previous questions, observations and suggestions by the reviewers and it seems all issues are resolved. I have a few of simple observations on this version, see below.
I recommend publication.

Experimental design

Initial problems in the description of the methods were resolved. Methods are not perfect (they never are), but are robust enough to have confidence in the results.

Validity of the findings

Results and findings are acceptable.

Additional comments

I have a few of simple observations on this version:

Line 31, add Weil et al. 2019 after Alvares-Filip et al. 2019 and to the reference list.
Line 91. a gloved hand I suppose??
Line 109:...... localized vs systemic in relation to the microbiome., but this may not be the case for the coral polyps and the colony. There is good evidence that some of the bacterial diseases have systemic effects (on reproduction and immune responses for example) at the colony level in scleractinian and octocorals (Petes et al. 2003; Weil et al. 2009; Mydlarz manuscripts).
Line 177-78. Did you observe any significant proliferation of new lesions in treated colonies vs. untreated (not touched) healthy controls? Do you think application of the pastes could elicit dispersion of pathogens to other areas of the colony, or to other colonies??

Maybe authors could consider including a short paragraph with the major limitations of the study before the conclusion.

---

## Round 0.3 · accepted · Accept

I am satisfied with the changes that have been made to the manuscript. With regards to the Weil et al 2019 reference, in my opinion, it can be used as a reference. Even though it is not part of the peer-reviewed literature, it is available online and therefore anyone can access it and will know that it is not a peer-reviewed document.